# Delexicalized and Minimally Supervised Parsing on Universal Dependencies

## Abstract

In this paper, we compare delexicalized transfer and minimally supervised parsing techniques on 32 different languages from Universal Dependencies treebank collection. The minimal supervision is in adding handcrafted universal grammatical rules for POS tags. The rules are incorporated into the unsupervised dependency parser in forms of external prior probabilities. We also experiment with learning this probabilities from other treebanks. The average attachment score of our parser is slightly lower then the delexicalized transfer parser, however, it performs better for languages from less resourced language families (non-Indo-European) and is therefore suitable for those, for which the treebanks often do not exist.

## 1 Introduction

In the last two decades, many dependency treebanks for various languages have been manually annotated. They differ in word categories (POS tagset), syntactic categories (dependency relations), and structure for individual language phenomena. The CoNLL shared tasks for dependency parsing (Buchholz and Marsi, 2006; Nivre et al., 2007) unified the file format, and thus the dependency parsers could easily work with 20 different treebanks. Still, the parsing outputs were not comparable between languages since the annotation styles differed even between closely related languages.

In recent years, there have been a huge effort to normalize dependency annotation styles. The Stanford dependencies (de Marneffe and Manning, 2008) were adjusted to be more universal across languages (de Marneffe et al., 2014). Mc-

donald et al. (2013) started to develop Google Universal Treebank, a collection of new treebanks with common annotation style using the Stanford dependencies and Universal tagset (Petrov et al., 2012) consisting of 12 part-of-speech tags. Zeman et al. (2012) produced a collection of treebanks HamleDT, in which about 30 treebanks were automatically converted to a Prague Dependency Treebank style (Hajič et al., 2006). Later, they converted all the treebanks also into the Stanford style (Rosa et al., 2014).

The researchers from the previously mentioned projects joined their efforts to create one common standard: Universal Dependencies (Nivre et al., 2016). They used the Stanford dependencies (de Marneffe et al., 2014) with minor changes, extended the Google universal tagset (Petrov et al., 2012) from 12 to 17 part-of-speech tags and used the Interset morphological features (Zeman, 2008) from the HamleDT project (Zeman et al., 2014). In the current version 1.2, Universal Dependencies collection (UD) consists of 37 treebanks of 33 different languages and it is very likely that it will continue growing and become common source and standard for many researchers. Now, it is time to revisit the dependency parsing methods and to investigate their behavior on this new unified style.

The goal of this paper is to apply cross language delexicalized transfer parsers (e.g. (McDonald et al., 2011)) on UD and compare their results with unsupervised and minimally supervised parser. Both the methods are intended for parsing languages, for which no annotated treebank exists and both the methods can profit from UD.

In the area of dependency parsing, the term "unsupervised" is understood as that no annotated treebanks are used for training and when supervised POS tags are used for grammar inference, we can deal with them only as with further un-

specified types of word.[1] Therefore, we introduce a minimally supervised parser: We use unsupervised dependency parser operating on supervised POS tags, however, we add external prior probabilities that push the inferred dependency trees in the right way. These external priors can be set manually as handwritten rules or trained on other treebanks, similarly as the transfer parsers. This allows us to compare the parser settings with different degrees of supervision:

- delexicalized training of supervised parsers

- minimally supervised parser using some external probabilities learned in supervised way

- minimally supervised parser using a couple of external probabilities set manually

- fully unsupervised parser

Ideally, the parser should learn only the language-independent characteristics of dependency trees. However, it is hard to define what such characteristics are. For each particular language, we will show what degree of supervision is the best for parsing. Our hypothesis is that a kind of minimally supervised parser can compete with delexicalized transfer parsers.

## 2 Related Work

There were many papers dealing with delexicalized parsing. Zeman and Resnik (2008) transfer a delexicalized parsing model to Danish and Swedish. McDonald et al. (2011) present a transfer-parser matrix from/to 9 European languages and introduce also multi-source transfer, where more training treebanks are concatenated to form more universal data. Both papers mention the problem of different annotation styles across treebanks, which complicates the transfer. Rosa (2015) uses already harmonized treebanks (Rosa et al., 2014) and compare the delexicalized parsing for Prague and Stanford annotation styles.

Unsupervised dependency parsing methods made a big progress started by the Dependency Model with Valence (Klein and Manning, 2004), which was further improved by many other researchers (Headden III et al., 2009; Blunsom and Cohn, 2010; Spitkovsky et al., 2011b; Spitkovsky

et al., 2012). Many of these works induce grammar based on the gold POS tags, some of them use unsupervised word classes (Spitkovsky et al., 2011a; Mareček, 2015). However, it seems that the research in this field declines in the recent years, probably because its results are still not able to compete with projection and delexicalized methods. Naseem et al. (2010) joined unsupervised grammar induction with a couple of syntactic rules.

## 3 Data

In all our experiments, we use the Universal Dependencies treebank collection[2] in its current version 1.2. For languages for which there is more than one treebank, we experiment only with the first one.[3] We also exclude 'Japan-KTC' treebank, since the full data are not available. Finally, we experiment with 32 dependency treebanks, each representing a different language. The treebanks, their language families, and their sizes are listed in Table 1.

Before training the parsers, all the treebanks are delexicalized. We substitute all the forms and lemmas by underscores, which are used for undefined values. The same is done with the morphological features and dependency relations. The only information remained is the universal POS tags and the dependency structure (the parent number for each token). The Universal Dependencies use POS tagset consisting of 17 POS tags listed in Table 2.

## 4 Experiments

In the following experiments, we compare delexicalized transfer parsing methods and minimally-supervised methods on the UD treebanks. All the experiments are conducted as if we parsed a language whose syntax is unknown for us. This means that we do not prefer training on syntactically similar languages, we do not prefer right branching or left branching, and do not add language specific word-order rules like preferring SVO or SOV, adjectives before nouns, prepositions vs. postpositions etc.

---

[1]In the fully unsupervised setting, we cannot for example simply push verbs to the roots and nouns to become their dependents. This is already a kind of supervision.

[2]universaldependencices.org

[3]We exclude 'Ancient Greek-PROIEL', 'Finnish-FTB', 'Japan-KTC', 'Latin-ITT', and 'Latin-PROIEL' treebanks.

| language | | family | tokens |
|---|---|---|---|
| ar | Arabic | Semitic | 282384 |
| bg | Bulgarian | Slavic | 156319 |
| cu | Old Slav. | Slavic | 57507 |
| cs | Czech | Slavic | 1503738 |
| da | Danish | Germanic | 100733 |
| de | German | Germanic | 293088 |
| el | Greek | Hellenic | 59156 |
| en | English | Germanic | 254830 |
| es | Spanish | Romance | 423346 |
| et | Estonian | Uralic | 6461 |
| eu | Basque | isolate | 121443 |
| fa | Persian | Iranian | 151624 |
| fi | Finnish | Uralic | 181022 |
| fr | French | Romance | 389764 |
| ga | Irish | Celtic | 23686 |
| got | Gothic | Germanic | 56128 |
| grc | Old Greek | Hellenic | 244993 |
| he | Hebrew | Semitic | 115535 |
| hi | Hindi | Indo-Iranian | 351704 |
| hr | Croatian | Slavic | 87765 |
| hu | Hungarian | Uralic | 26538 |
| id | Indonesian | Malayic | 121923 |
| it | Italian | Romance | 252967 |
| la | Latin | Romance | 47303 |
| nl | Dutch | Germanic | 200654 |
| no | Norwegian | Germanic | 311277 |
| pl | Polish | Slavic | 83571 |
| pt | Portuguese | Romance | 212545 |
| ro | Romanian | Romance | 12094 |
| sl | Slovenian | Slavic | 140418 |
| sv | Swedish | Germanic | 96819 |
| ta | Tamil | Dravidian | 9581 |

Table 1: Languages and their families used in the experiments and sizes of the respective treebanks.

| ADJ | adjective | PART | particle |
|---|---|---|---|
| ADP | adposition | PRON | pronoun |
| ADV | adverb | PROPN | proper noun |
| AUX | auxiliary verb | PUNCT | punctuation |
| CONJ | coord. conj. | SCONJ | subord. conj. |
| DET | determiner | SYM | symbol |
| INTJ | interjection | VERB | verb |
| NOUN | noun | X | other |
| NUM | numeral | | |

Table 2: List of part-of-speech tags used in Universal-Dependencies treebanks.

## 4.1 Delexicalized parsing

We apply the multi-source transfer of delexicalized parser on the UD treebanks in a similar way as McDonald et al. (2011). We use the leave-one-out method: for each language, the delexicalized parser is trained on all other treebanks excluding the one on which the parser is tested. Since all the treebanks share the tagset and annotation style, the training data can be simply concatenated together. To decrease the size of the training data and to reduce the training time, we decided to take only first 10,000 tokens for each language, so the final size of the training data is about 300,000 tokens, which is enough for training delexicalized parser. We use the Malt parser[4] (Nivre, 2009), and MST parser (McDonald et al., 2005) with several parameter settings. The results are shown in Table 5.

## 4.2 Minimally supervised parsing

The goal of this paper is to investigate whether the unsupervised parser with added external prior probabilities reflecting the universal annotation scheme is able to compete with the delexicalized methods described in Section 4.1.

We use the unsupervised dependency parser (UDP) implemented by Mareček and Straka (2013). The reason for this choice was that it has reasonably good results across many languages (Mareček, 2015), the source code is freely available,[5] and because it includes a mechanism how to import external probabilities. The UDP is based on Dependency Model with Valence, a generative model which consists of two sub-models:

- Stop model $p_{stop}(\cdot|t_g, dir)$ represents probability of not generating another dependent in direction $dir$ to a node with POS tag $t_g$. The direction $dir$ can be left or right. If $p_{stop} = 1$, the node with the tag $t_g$ cannot have any dependent in direction $dir$. If it is 1 in both directions, the node is a leaf.

- Attach model $p_{attach}(t_d|t_g, dir)$ represents probability that the dependent of the node with POS tag $t_g$ in direction $dir$ is labeled with POS tag $t_d$.

In other words, the *stop* model generates edges, while the *attach* model generates POS tags for the

---

[4]Malt parser in the current version 1.8.1 (http://maltparser.org)

[5]http://ufal.mff.cuni.cz/udp

| $t_g$ | $p_{stop}^{ext}$ |
|---|---|
| ADP, ADV, AUX, CONJ, DET, INTJ, NUM, PART, PRON, PUNCT, SCONJ, SYM | 1.0 |
| ADJ | 0.9 |
| PROPN | 0.7 |
| X | 0.5 |
| NOUN | 0.3 |
| VERB | 0.1 |

Table 3: Manual assignment of *stop* probabilities for individual POS tags.

new nodes. The inference is done using blocked Gibbs sampling (Gilks et al., 1996). During the inference, the *attach* and the *stop* probabilities can be combined linearly with external prior probabilities $p^{ext}$:

$$p_{stop}^{final} = (1 - \lambda_{stop}) \cdot p_{stop} + \lambda_{stop} \cdot p_{stop}^{ext},$$

$$p_{attach}^{final} = (1 - \lambda_{atach}) \cdot p_{attach} + \lambda_{attach} \cdot p_{attach}^{ext},$$

where the parameters $\lambda$ define their weights. In the original paper (Mareček and Straka, 2013), the external priors $p_{stop}^{ext}$ were computed based on the reducibility principle on a big raw corpora.

### 4.2.1 Manually Assigned Priors

We use the external prior probabilities to define grammatical rules for POS tags based on UD annotation style. The first type of priors describes how likely a node with a particular POS is a leaf. We manually set the $p_{stop}^{ext}$ as listed in Table 3. Even though it is possible to define different left and right $p_{stop}^{ext}$, we decided to set it equally for both the directions, since it is linguistically more language independent.

In a similar way, we predefine external priors for $p_{attach}^{ext}$, describing dependency edges.[6] Preliminary experiments showed that less is more in this type of rules. We ended up only with four rules for attaching punctuation and prepositions, as defined in Table 4.[7] Similarly as for $p_{stop}^{ext}$, we set them equally for both left and right directions. We set $\lambda_{attach} = 0$ for all other possible types of edges, since the priors are not defined for them.

---

[6]We had to change the original parser code to do this.
[7]Note that for example $p_{attach}^{ext}(PUNC|VERB, dir) = 1$ does not mean that all the dependents of VERB must be PUNC. Since the $\lambda_{attach}$ is less than one, the value 1 only pushes punctuation to be attached below verbs.

| $t_g$ | $t_d$ | $p_{attach}^{ext}$ |
|---|---|---|
| VERB | PUNCT | 1.0 |
| NOUN | PUNCT | 0.0 |
| VERB | ADP | 0.0 |
| NOUN | ADP | 1.0 |

Table 4: Manual assignment of *attach* probabilities for some types of edges.

### 4.2.2 Automatically Assigned Priors

Instead of setting the external probabilities manually, we can compute them automatically from other treebanks. Such experiments are somewhere in the middle between delexicalized parsers and the minimally supervised parser with some manually added knowledge. They learn some regularities but not as many as the delexicalized parsers do.

Similarly as for delexicalized transfer parser, we compute the probabilities on all treebanks but the one which is currently tested. The probabilities are computed in the following way:

$$p_{stop}^{ext}(\cdot|t_g, dir) = \frac{NC(t_g)}{CC(t_g, dir) + NC(t_g)},$$

where $NC(t_g)$ is count of all nodes labelled with tag $t_g$ across all the training treebanks, $CC(t_g, dir)$ is the total number of children in direction $dir$ of all $t_g$ nodes in the treebanks, and

$$p_{attach}^{ext}(t_d|t_g, dir) = \frac{NE(t_g, t_d, dir)}{NE(t_g, *, dir)},$$

where $NE(t_g, t_d, dir)$ is number of dependency edges where the governing node has the POS tag $t_g$, and the dependent node $t_d$ and is in direction $dir$ from the governing one.

We introduce two additional experiments: *direction-dependent learned priors* (DDLP) and *direction-independent learned priors* (DILP). The external probabilities for DDLP are computed according to the previously mentioned formulas.

In DILP, the probabilities are independent on the direction parameter $dir$. $p_{stop}^{ext}(\cdot|t_g)$ and $p_{attach}^{ext}(t_d|t_g)$ obtain the same values for both directions. Such approach is therefore less supervised. We suppose, that it gains worse results form majority of languages, however, it could be better for some of languages with word ordering different from the majority of languages.

| lang. | MST parser | | Malt parser | | UDP | | | |
|---|---|---|---|---|---|---|---|---|
| | proj | nproj | lazy | nivre | basic | +rules | DDLP | DILP |
| ar | 48.8 | 51.2 | 50.2 | 50.4 | 42.9 | 51.7 | **55.2** | 48.0 |
| bg | **79.0** | 78.5 | 78.1 | 77.4 | 52.6 | 74.6 | 73.2 | 66.8 |
| cu | 64.8 | **66.0** | 63.1 | 62.6 | 46.8 | 58.1 | 64.5 | 59.7 |
| cs | 68.0 | **68.4** | 66.3 | 65.8 | 43.6 | 60.2 | 62.8 | 55.4 |
| da | 71.0 | 71.7 | 66.9 | 67.2 | 40 9 | 57.7 | **89.6** | 54.8 |
| de | 69.8 | **70.0** | 65.2 | 65.4 | 37.4 | 60.9 | 63.5 | 59.8 |
| el | 64.3 | **64.9** | 63.8 | 64.1 | 13.1 | 63.2 | 62.3 | 55.9 |
| en | 62.1 | **62.4** | 58.2 | 58.3 | 28.1 | 54.6 | 54.5 | 53.0 |
| es | 71.5 | **72.2** | 68.8 | 69.0 | 20.4 | 63.7 | 66.3 | 56.1 |
| et | 76.4 | 75.1 | 70.9 | 70.5 | 26.8 | 79.2 | 74.6 | **80.3** |
| eu | 50.0 | 51.2 | 51.8 | 50.9 | 47.1 | **53.8** | 50.5 | 52.3 |
| fa | 52.8 | 54.8 | 54.0 | 54.0 | 41.0 | 54.8 | **57.5** | 45.0 |
| fi | 55.1 | **55.8** | 50.5 | 50.4 | 27.6 | 48.8 | 46.6 | 48.7 |
| fr | 74.3 | **74.8** | 71.5 | 71.5 | 36.0 | 65.8 | 69.0 | 57.9 |
| ga | 60.7 | 61.4 | 61.1 | 61.3 | 37.1 | 60.2 | **61.5** | 57.3 |
| got | 63.6 | **64.5** | 62.8 | 62.1 | 47.3 | 60.2 | 62.4 | 57.4 |
| grc | 47.2 | 48.0 | 45.8 | 45.5 | 41.2 | 50.6 | **51.4** | 51.2 |
| he | 62.5 | **64.0** | 63.1 | 62.7 | 28.2 | 62.4 | **64.0** | 56.5 |
| hi | 33.5 | 34.2 | 35.5 | 35.1 | 42.3 | 50.9 | 38.4 | **54.0** |
| hr | 69.3 | **69.4** | 67.3 | 67.1 | 24.7 | 61.5 | 63.4 | 54.8 |
| hu | 57.4 | 58.0 | 54.6 | 54.2 | 53.4 | 57.4 | 55.4 | **62.8** |
| id | 58.5 | 61.0 | 59.2 | 58.6 | 22.7 | 48.4 | **61.3** | 51.6 |
| it | 76.4 | **77.1** | 74.0 | 73.8 | 42.3 | 68.8 | 71.5 | 60.1 |
| la | **56.5** | 55.9 | 55.5 | 55.8 | 47.0 | 51.8 | 52.0 | 47.1 |
| nl | **60.2** | 60.1 | 56.5 | 57.3 | 37.5 | 51.2 | 54.9 | 48.5 |
| no | 70.2 | **70.4** | 67.2 | 66.9 | 40.9 | 58.5 | 61.4 | 55.7 |
| pl | 75.6 | **76.0** | 74.7 | 75.0 | 63.8 | 68.0 | 67.7 | 64.6 |
| pt | 73.9 | **74.3** | 72.4 | 71.7 | 40.1 | 64.6 | 69.4 | 58.2 |
| ro | 68.3 | **69.3** | 68.2 | 67.7 | 60.4 | 57.9 | 66.3 | 58.9 |
| sl | 72.2 | **72.8** | 71.2 | 70.6 | 48.6 | 68.6 | 64.9 | 56.8 |
| sv | 70.2 | **70.8** | 66.2 | 66.2 | 41.5 | 59.5 | 61.7 | 58.7 |
| ta | 34.3 | 36.5 | 35.5 | 35.6 | 52.2 | 52.9 | 48.4 | **58.4** |
| avg | 63.1 | **63.8** | 61.6 | 61.4 | 39.9 | 59.4 | 60.5 | 56.5 |

Table 5: Unlabeled attachment scores for the parsers across the languages. The best results are in bold. For MST parser, we used the second order features and its projective (*proj*) and non-projective (*non-proj*) variant. For the Malt parser, we used lib-SVM training and stacklazy (*lazy*) and nivreeager (*nivre*) algorithms. Unsupervised dependency parser (*UDP*) was tested without any external priors (*basic*), with manual prior probabilities (*+rules*), and with automatically learned probabilities direction dependent (*DDLP*) and direction independent (*DILP*).

## 5 Results

The results of delexicalized transfer parsers, unsupervised parser and minimally supervised parsers with different degrees of supervision on Universal Dependencies are compared in Table 5. We try several settings of parameters for both Malt parser and MST parser, and show the results of two of them for each one.[8] We run the Unsupervised dependency parser by Mareček and Straka (2013), labeled as *UDP*. For UDP, we report four different settings. The *basic* variant is completely unsupervised parsing without any external prior probabilities. The *+rules* column shows the results of

---

[8]The results of different parameter settings for both parser varied only little (at most 2% difference for all the languages).

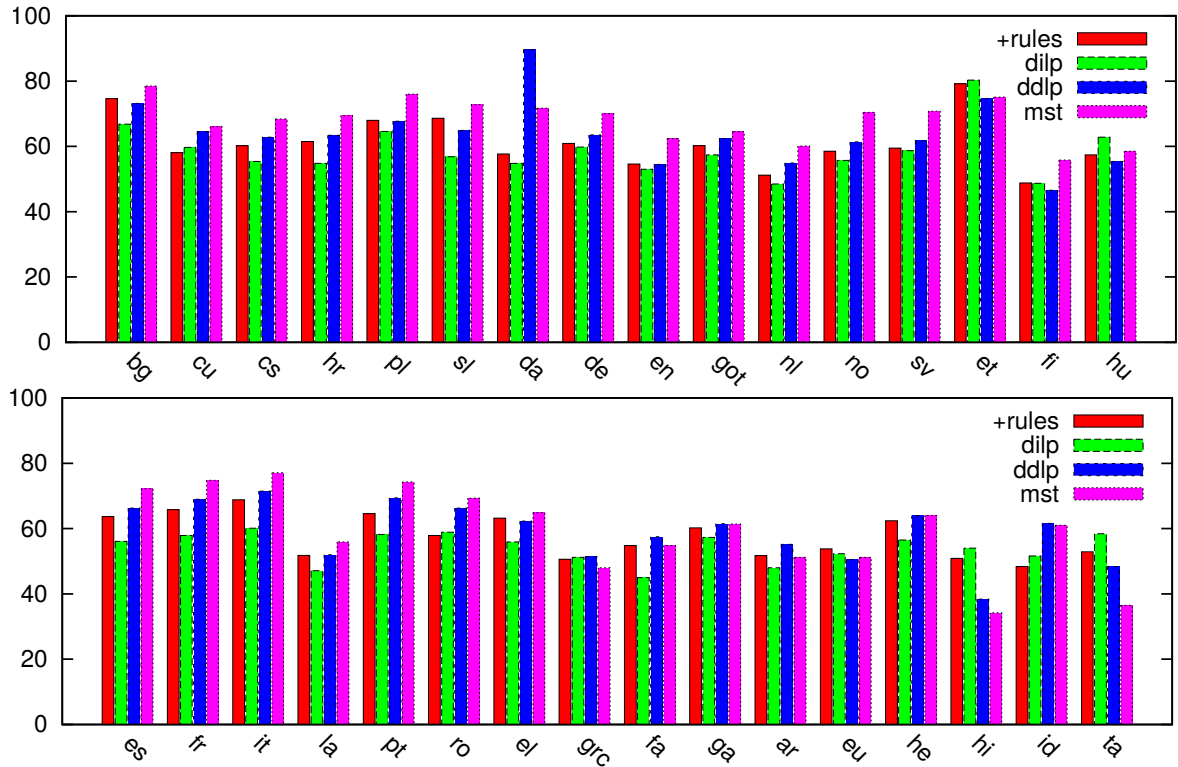

Figure 1: Comparison of delexicalized parsing methods with different degrees of supervision. UDP with manually set priors (+rules), direction dependent (DDLP) and independent (DILP) learning of priors versus delexicalized transfer of MST parser (mst). Languages are ordered according to their language families: Slavic (bg, cu, cs, hr, pl, sl), Germanic (da, de, en, got, nl, no, sv), Romance (es, fr, it, la, pt, ro), Hellenic (el, grc), Uralic (et, fi, hu), and others (fa, ga, ar, eu, he, hi, id, ta).

our minimally supervised parser (Section 4.2.1) using the external probabilities defined manually (Tables 3 and 4). Both the $\lambda_{stop}$ and $\lambda_{attach}$ parameters are set to 0.5. The *DDLP* and *DILP* variants use automatically learned prior probabilities form other treebanks (Section 4.2.2).

## 5.1 Discussion

It is evident that the MST parser achieved the best scores. It parsed best 20 out of 32 languages and its non-projective variant reached 63.8% averaged attachment score. The Malt parser was worse than MST by 2% in the averaged attachment score.[9] The basic UDP without additional rules performs very poorly, however, with added external prior probabilities, it is competitive with the delexicalized transfer parser methods. 12 out of 32 languages were parsed better by UDP using one variant of the external priors.

With hand-written prior probabilities (+*rules*),

the averaged attachment score reached only 59%, however, it is better than the MST parser on 6 languages: Arabic, Estonian, Basque, Old Greek, Hindi, and Tamil, in two cases by a wide margin. For Persian, the scores are equal.

The averaged attachment score for UDP with direction-independent learned priors (DILP) is even lower (56.5%), however, it parsed 6 languages better than MST: Estonian, Basque, Old Greek, Hindi, Hungarian, and Tamil. Direction dependent learning of priors end up with 60.5% attachment score and 9 languages better than MST.

Based on these results, we can say that the minimally supervised parser, which takes less information from other annotated treebanks, is more suitable for the more exotic languages, i.e. for languages whose families are less common among the annotated treebanks. Figure 4.2.2 shows histograms of attachment scores across languages, now ordered according to the language families. All the Slavic and Romance languages and almost all the Germanic languages[10] are parsed best by

---

[9]We used the Malt parser with its default feature set. Tuning in this specific delexicalized task would probably bring a bit better results.

[10]Danish is the only exception.

the MST parser. Finnish from the three Uralic languages and Greek from the two Hellenic languages are also parsed best by MST. Other 12 languages were better parsed by one of the less supervised methods.

Less-resourced languages, for which the annotated treebanks are missing, may be therefore better parsed by less supervised parsers, especially if they do not belong to the Indo-European language family. The MST transfer parser has probably been over-trained on these Indo-European family languages and is not able to generalize enough to more distant languages. The rules we added to the unsupervised dependency parser (+*rules* experiment) are universal in the direction of dependencies (left/right branching) and cover much more languages.

## 5.2 Transfer parser comparison between different styles

We compare the best transfer parser results also with the previous works. Even though the results are not directly comparable, because different annotation styles were used, we suppose that the annotation unification across the treebanks in Universal Dependencies should improve the transfer parser scores. McDonald et al. (2011) presented 61.7% of averaged accuracy over 8 languages. On the same languages, our transfer parser on UD reached 70.1%. When compared to Rosa (2015), we experimented with 23 common languages, our average score on them is 62.5%, Rosa's is 56.6%. The higher attachment scores in our experiments confirms that the annotations in UD treebanks are more unified and serve better for transferring between languages.

## 6 Conclusions

We used the Universal Dependencies treebank collection to test delexicalized transfer parsers and unsupervised dependency parser enriched by external *attach* and *stop* prior probabilities. We found that whereas the MST delexicalized transfer parser is better in average, our minimally supervised parser performs better on many non-Indo-European languages and therefore can be suitable to parse often low-resourced exotic languages, for which treebanks do not exist.

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
