# Peer review of "Delexicalized and Minimally Supervised Parsing on Universal Dependencies"

_CoNLL 2016 — decision unknown_

[Official Review · Reviewer 1 · rating 2 · confidence 4]
soundness 4 · originality 3 · clarity 4 · impact 2 · substance 2 · appropriateness 5 · meaningful comparison 3 · replicability 4 · presentation format Poster

This paper presents results on the UD treebanks to test delexicalized transfer
parsers and an unsupervised parser which is enriched with external
probabilities.

The paper is interesting, but I think it could be improved further.

(5.2) "McDonald et al. (2011) presented 61.7% of averaged accuracy over 8
languages. On the same languages, our transfer parser on UD reached 70.1%."
Mcdonald et al could not use the UD treebanks since they were not available,
you should definitely state that this is the case here.

In footnote 9 you say: "We used the Malt parser with its default feature set.
Tuning in this specific delexicalized task would probably bring a
bit better results." You are using MaltParser with default settings, why don't
you use MaltOptimizer? Optimizing one model would be very easy. 
In the same way MSTParser could be optimized further.
In the same line, why don't you use more recent parsers that produce better
results? These parsers have been already applied to universal dependencies with
the leave one out setup (see references below). For instance, the authors say
that  the unsupervised parser "performs better for languages from less
resourced language families (non-Indo-European)", it would be interesting to
see whether this holds with more recent (and cross lingual) parsers.

Probabilities: Why do you use this probabilities? it seems like a random
decision (Tables 3-4) (esp 3), at least we need more details or a set of
experiments to see whether they make sense or not.

There are some papers that the authors should take into account.

1. Cross-Lingual Dependency Parsing with Universal Dependencies and Predicted
PoS Labels
J Tiedemann
2. One model, two languages: training bilingual parsers with harmonized
treebanks
D Vilares, MA Alonso, C GÃ³mez-RodrÃ­guez  (it presents results with
MaltParser)

And for results with more recent parsers (and also delexicalized parsers):
1. Crosslingual dependency parsing based on distributed representations. 
Jiang Guo, Wanxiang Che, David
Yarowsky, Haifeng Wang, and Ting Liu. 2015.  In Proc. of ACL

2. Many languages, one parser
W Ammar, G Mulcaire, M Ballesteros, C Dyer, NA Smith

-Minor points:
 I don't think we need Table 1 and Table 2, this could be solved with a
footnote to the UD website. Perhaps Table 2 should be included due to the
probabilities, but Table 1 definitely not.

[Official Review · Reviewer 2 · rating 3 · confidence 4]
soundness 3 · originality 2 · clarity 4 · impact 2 · substance 4 · appropriateness 5 · meaningful comparison 4 · replicability 3 · presentation format Poster

This paper evaluates a minimally supervised dependency parser -- a version of
the DMV model with manually set prior probabilities -- on (most of) the
treebanks from Universal Dependencies, v1.2. It reports results that are on
average slightly lower than a couple of delexicalized transfer parsers but
(sometimes substantially) better on a few non-Indo-European languages.

The idea of biasing an otherwise unsupervised parser with some basic
"universal" rules have been used a number of times before in the literature, so
the main value of the present paper is an empirical evaluation of this approach
on the new UD treebanks. However, the approach and evaluation leaves some 
questions unanswered.

First of all, I want to know why only unlabeled parsing is considered. This may
have been appropriate (or at least necessary) before dependency labels were
standardised, but the whole point of UD is to give a uniform analysis in terms
of typed dependencies, and any parsing approach that does not take this into
account seems misguided. And since the approach is based on manually defined
universal rules, it would have been easy enough to formulate rules for labeled
dependencies.

Second, I would like to know more about how the prior probabilities were set
or, in other words, what universal grammar they are meant to encode and how.
Were alternatives tested and, if so, how were they evaluated? In the present
version of the paper, we are just presented with a bunch of numbers without any
explanation or justification except that they are âbased on UD annotation
styleâ.

Third, one of the main claims of the paper is that the unsupervised system
works better for non-Indo-European languages. This seems to be supported by the
raw numbers, but what exactly is going on here? What types of dependencies are
handled better by the unsupervised system? Even though a full error analysis
would be out of scope in a short paper, an analysis of a small sample could be
really interesting.

Finally, the comparison to the delexicalized transfer parsers seems to be
biased by a number of factors. Restricting it to unlabeled dependencies is one
such thing, since the delexicalized parser could easily have produced labeled
dependencies. Another thing is the amount of training data, which was
arbitrarily restricted to 10,000 tokens per treebank. Finally, it seems that
the delexicalized parsers were not properly tuned. Just replacing word forms
and lemmas by underscores without revising the feature models is not likely to
produce optimal results.